# DNA Vaccine Encoding a Modified Hemagglutinin Trimer of Avian Influenza A Virus H5N8 Protects Mice from Viral Challenge

**DOI:** 10.3390/vaccines12050538

**Published:** 2024-05-14

**Authors:** Victoria R. Litvinova, Andrey P. Rudometov, Nadezhda B. Rudometova, Denis N. Kisakov, Mariya B. Borgoyakova, Lyubov A. Kisakova, Ekaterina V. Starostina, Anastasia A. Fando, Vladimir A. Yakovlev, Elena V. Tigeeva, Ksenia I. Ivanova, Andrei S. Gudymo, Tatiana N. Ilyicheva, Vasiliy Yu. Marchenko, Artemiy A. Sergeev, Alexander A. Ilyichev, Larisa I. Karpenko

**Affiliations:** Federal Budgetary Research Institution State Research Center of Virology and Biotechnology «Vector», Rospotrebnadzor, Koltsovo 630559, Novosibirsk Region, Russia; litvinova_vr@vector.nsc.ru (V.R.L.); andreeva_nb@vector.nsc.ru (N.B.R.); def_2003@mail.ru (D.N.K.); borgoyakova_mb@vector.nsc.ru (M.B.B.); orlova_la@vector.nsc.ru (L.A.K.); starostina_ev@vector.nsc.ru (E.V.S.); fando_aa@vector.nsc.ru (A.A.F.); tigeeva_ev@vector.nsc.ru (E.V.T.); ivanova_ki@vector.nsc.ru (K.I.I.); gudymo_as@vector.nsc.ru (A.S.G.); ilicheva_tn@vector.nsc.ru (T.N.I.); marchenko_vyu@vector.nsc.ru (V.Y.M.); sergeev_aa@vector.nsc.ru (A.A.S.); ilyichev@vector.nsc.ru (A.A.I.); karpenko@vector.nsc.ru (L.I.K.)

**Keywords:** influenza A (H5N8) virus, DNA vaccines, immunogenicity, jet injection

## Abstract

The development of a safe and effective vaccine against avian influenza A virus (AIV) H5N8 is relevant due to the widespread distribution of this virus in the bird population and the existing potential risk of human infection, which can lead to significant public health concerns. Here, we developed an experimental pVAX-H5 DNA vaccine encoding a modified trimer of AIV H5N8 hemagglutinin. Immunization of BALB/c mice with pVAX-H5 using jet injection elicited high titer antibody response (the average titer in ELISA was 1 × 10^5^), and generated a high level of neutralizing antibodies against H5N8 and T-cell response, as determined by ELISpot analysis. Both liquid and lyophilized forms of pVAX-H5 DNA vaccine provided 100% protection of immunized mice against lethal challenge with influenza A virus A/turkey/Stavropol/320-01/2020 (H5N8). The results obtained indicate that pVAX-H5 has good opportunities as a vaccine candidate against the influenza A virus (H5N8).

## 1. Introduction

Avian influenza virus (AIV) subtype A (H5N8) circulates among wild birds and poultry, causing their mortality and periodically infecting mammals [1]. H5N8 viruses of clade 2.3.4.4a were first identified in China and South Korea in early 2014, and then they spread to other countries in Asia and Europe, and by the end of 2014 reached North America [2]. The first reports of AIV 2.3.4.4b H5N8 viruses came from China in May 2016 [3]. Subsequently, AIV H5N8 viruses of this lineage quickly spread through migratory birds to Asia, Europe, and Africa, causing the death of wild and domestic bird populations in many countries [4,5]. In 2020, cases of human infection with the influenza A virus (H5N8) were identified; later, this strain was recommended by WHO as a vaccine strain [6,7]. Thus, the evolution of AIV continues, indicating its high pandemic potential. As part of the influenza pandemic preparedness program, the World Health Organization (WHO) calls for increased efforts to ensure emergency preparedness [8], in particular the development of vaccines against avian influenza viruses.

The fight against SARS-CoV-2 has stimulated the development of new vaccine platforms; the most promising among them are vaccines based on nucleic acids—mRNA and DNA vaccines [9]. These vaccines offer several very important advantages over other types of vaccines, including stimulation of both humoral and T cell responses, the speed of the vaccine development, and the relative ease and universality of large-scale production. Vaccines based on nucleic acids do not cause an unwanted anti-vector immune response, which is typical for vaccines based on viral vectors and, therefore, can be administered repeatedly. DNA vaccines are also stable over a wide temperature range [10,11]. So, vaccines based on nucleic acids can be considered as a platform technology that makes it quite easy to replace the target immunogen, without the need to change production [12,13]. The first DNA vaccine approved for human vaccination was the ZyCoV-D vaccine, designed to prevent COVID-19, developed by the Indian company Zydus Lifesciences Limited [14,15]. Many candidate preventive DNA vaccines against human viral and bacterial diseases [16] show that DNA vaccines are safe and have an acceptable reactogenicity profile [16,17,18]. A number of DNA vaccines against AIV are being developed [19,20]. A DNA vaccine encoding hemagglutinin of the influenza A virus (H5N1) successfully passed phase I of clinical trials [21]. It should also be noted that the DNA vaccine against highly pathogenic influenza A (H5N1) for chicken immunization, developed by Agrilabs, has recently been licensed by the US Department of Agriculture [22]. However, a number of issues related to this platform remain not fully resolved, in particular problems associated with the delivery of a DNA vaccine [11,23,24].

To produce target immunogens, the DNA vaccine has to get inside the cell from the intercellular space, breaking through two membranes—cellular and nuclear, which occurs with low efficiency without the use of agents that increase the transfection properties of DNA [25]. To increase the immunogenicity of DNA vaccines, a number of methods are used, both physical (electroporation (EP) [26], gene gun [27], microneedles [28], jet injection [29]) and chemical (liposomes, dendrimers, etc.), as well as molecular and traditional adjuvants. A relatively new, but already proven method of delivering DNA vaccines is the jet injection method using individual nozzles [30,31,32,33]. Prophylactic DNA vaccines delivered via needle-free injection systems (NFISs) induce immunogenicity comparable to electroporation with minimal adverse effects on the patient [34,35,36,37]. Therefore, in our research, we focused on this method of delivering a DNA vaccine.

HA is one of the main components of influenza vaccines, since it induces the formation of neutralizing antibodies, which ensure the formation of protective immunity to the virus [38,39]. Influenza virus HA in its native form exists as a trimer on the viral membrane; this fact must be considered in the design of DNA vaccines encoding the HA gene. Since the conformation of the influenza virus HA protein is important for its immunogenicity, in our study, we modified the HA trimer by introducing point mutations in the region of the pH switches, based on data [40,41].

Here, we described the design and development of a DNA vaccine encoding a modified hemagglutinin of highly pathogenic avian influenza A virus H5N8 and the evaluation of its immunogenic and protective properties.

## 2. Materials and Methods

### 2.1. Strains of Viruses, Bacteria, Cell Cultures

The influenza virus strain A/turkey/Stavropol/320-01/2020 (H5N8) (EPI_ISL_1114749) was used in the virus neutralization assay; the homologous strain isolated from humans A/Astrakhan/3212/2020 (H5N8) (EPI_ISL_1038924) was used for challenge experiments.

*E. coli* Stbl3 strain (F′ proA+B+ lacIq ∆(lacZ)M15 zzf::Tn10 (TetR) ∆(ara-leu) 7697 araD139 fhuA ∆lacX74 galK16 galE15 e14- Φ80dlacZ∆M15 recA1 relA1 endA1 nupG rpsL (StrR) rph spoT1 ∆(mrr-hsdRMS-mcrBC)) was used to produce plasmid DNA.

Human Embryonic Kidney 293 (HEK293) and Madin-Darby canine kidney (MDCK) cell cultures (Cell Culture Collection of FBRI SRC VB «Vector», Rospotrebnadzor) were used in the work.

### 2.2. Design of DNA Vaccine pVAX-H5

The design of the nucleotide sequence encoding hemagglutinin of the influenza A virus (H5N8) was carried out on the basis of the native hemagglutinin gene (A/turkey/Stavropol/320-01/2020 EPI_ISL_1114749). The sequence encoding the stabilized hemagglutinin protein was codon-optimized for expression in mammalian cells using the Codon Adaptation Tool [42] and then synthesized by the commercial company DNA-Synthesis (Russia). Next, the synthesized gene was cloned into the pVAX1 vector (Thermo Fisher Scientific, Waltham, MA, USA) using the AsuNHI and CciNI restriction sites (SibEnzyme, Novosibirsk, Russia). The structure of the resulting recombinant plasmid pVAX-H5 was confirmed by Sanger sequencing and restriction analysis. Modeling of the spatial structure of the designed HA molecule was carried out as described previously [43] in the Alphafold2 colab program version 1.5.5 [44] and models were visualized with RCSB PDB website, Mol*Plagin 3.43.1 3D Viewer tool [45].

### 2.3. Production of DNA Vaccine pVAX-H5

Competent *E. coli* Stbl3 cells were transformed with the pVAX-H5 plasmid. Recombinant *E. coli* Stbl3 cells were cultivated at 37 °C in LB medium with kanamycin (100 μg/mL). For analytical purposes, plasmid DNA was isolated from bacterial cells using the commercial “Hipure Plasmid Mini Kit” (Magen, Guangzhou, China) according to the manufacturer’s recommendations. For mice immunization, plasmid DNA was isolated using the EndoFree Plasmid Giga Kit (Qiagen, Hilden, Germany) and dissolved in saline. Quantitative and qualitative assessment of isolated plasmid DNA was carried out using a NanoDrop™ OneC spectrophotometer (Thermo Fisher Scientific, USA) at wavelengths of 260, 280, and 230 nm. Endotoxin detection was carried out using the LAL reagent Endosafe-PTS (Charles River Laboratories, Wilmington, MA, USA) according to the manufacturer’s instructions.

### 2.4. Obtaining of Lyophilized Vaccine Preparation

A solution of plasmid DNA in saline containing 2 mg/mL sucrose was transferred into cryo-tubes and frozen in liquid nitrogen for 10 min. Then, the tubes with the frozen solution were transferred to a pre-cooled freeze-drying chamber Alpha 1–4 LD plusTM (Martin Christ Gefriertrocknungsanlagen GmbH, Osterode am Harz, Germany) and dried for 48 h at 0.04 mbar until the ice condenser temperature reached −50 °C and 12 h at 0.001 mbar until the ice condenser temperature reached −80 °C.

### 2.5. Analysis of the Hemagglutinin Gene Expression Using RT-PCR

Transfection of the HEK293 cell line with the recombinant plasmid pVAX-H5 was carried out using Lipofectamin (Lipofectamin 3000, Thermo Fisher Scientific, USA) in a 12-well plate format according to the method recommended by the manufacturer. A total of 48 h after transfection, cells were collected by centrifugation. After cell sedimentation, the culture medium was collected and frozen at −20 °C for subsequent use in immunoblotting. The cell sediment was suspended in PBS and the total RNA pool was isolated using the LIRA reagent (Biolabmix, Novosibirsk, Russia) according to the manufacturer’s recommendations. Isolated RNA samples were treated with DNase (Thermo Fisher Scientific, USA) and stored at −80 °C.

The reverse transcription reaction was carried out using primers (Table 1) to obtain cDNA of the gene encoding hemagglutinin of the influenza A virus (H5N8) using the BioMaster RT-PCR-Extra kit (2×) (Biolabmix, Russia) according to the manufacturer’s instructions. Primers were synthesized by the commercial company Biosset (Russia). After RT-PCR, amplification products were analyzed using electrophoresis in a 1% agarose gel.

### 2.6. Western Blot Analysis

Electrophoretic separation of proteins was carried out using Laemmli PAGE in a discontinuous buffer system in the presence of 0.1% SDS in Tris-glycine buffer. Then, the proteins of cell lysates and culture medium, separated electrophoretically in a polyacrylamide gel, were transferred to a nitrocellulose membrane (Cytiva, Marlborough, MA, USA). Protein transfer was controlled by staining the membrane with crimson C for several minutes, followed by washing the dye with distilled water.

Western blot analysis was performed using the SNAP i.d system 2.0 (Merck Millipore, Burlington, MA, USA) in accordance with the manufacturer’s recommendations. The serum of a ferret immunized with influenza A (H5N8) virus (1:200 dilution) was used as primary antibodies (FBRI SRC VB «Vector», Rospotrebnadzor). A fraction of anti-ferret immunoglobulins from mouse serum (dilution 1:3000) (FBRI SRC VB «Vector», Rospotrebnadzor) was used as secondary antibodies. Immune complexes were visualized using anti-mouse antibodies conjugated to alkaline phosphatase (1:5000) (Sigma-Aldrich, Saint Louis, MO, USA) and subsequent incubation with phosphatase substrate 1-Step™ NBT/BCIP (Thermo Fisher Scientific, USA).

### 2.7. Immunization of BALB/c Mice with DNA Vaccine pVAX-H5

Work with animals was carried out in accordance with the Guide for the Care and Use of Laboratory Animals. The protocols were approved by the Animal Care and Use Committee (IACUC) at the State Research Center of Virology and Biotechnology “Vector” (BEC Protocol No. 1 of 03/21/2023).

For immunization, female BALB/c mice weighing 16–18 g were used. DNA vaccine preparations dissolved in 50 μL of physiological solution were injected into the area of the left hind paw by jet injection using a Comfort-IN injector (Australia) with individual nozzle attachments. The drug administration parameters were as follows: jet speed 220 m/s, pressure 6.5 bar, injection time 0.33 s. To immobilize the animals, inhalation anesthesia (RWD Life Science, Shenzhen, China) with a 2.5% isoflurane solution was used for 4–5 min. The hair was removed from the thigh using depilatory gel, followed by an injection procedure.

The immunization experiments were divided into two stages. At the first stage, the humoral and cellular responses of the experimental DNA vaccine were assessed. The animals were divided into three groups (6 animals in each group). Group 1 was injected with 100 μg of pVAX-H5 solution in saline (pVAX-H5); Group 2 was injected 100 μg of pVAX1 (pVAX); Group 3 consisted of intact animals (intact). Immunization was carried out twice with an interval of 21 days. A total of 14 days after the second immunization, blood was collected from the retrobulbar sinus of the eye to assess the humoral response and spleens were explanted to assess the T-cell response. To collect the spleens, animals were removed from the experiment using the cervical dislocation method.

At the second stage, animals were immunized to evaluate the protective properties of the developed DNA vaccine. The animals were divided into five groups (10 animals in each group). Group 1 was injected with 100 μg of pVAX-H5 solution in saline (pVAX-H5); Group 2 was injected with 100 μg of lyophilized pVAX-H5 dissolved in pyrogen-free water (pVAX-H5 lyoph, sucrose); Group 3 was injected with 25 μL of inactivated influenza A virus (H5N8) intramuscularly in combination with incomplete Freud’s adjuvant (1 × 10^6^ viral particles per mL) in a total volume of 100 μL/individual (inactivated H5N8); Group 4 was injected with 100 μg of pVAX1 in saline (pVAX); Group 5 consisted of intact animals (intact). Immunization was carried out twice with an interval of 21 days. A total of 14 days after the second immunization, blood was collected from the retro-orbital sinus.

### 2.8. Enzyme-Linked Immunosorbent Assay (ELISA)

ELISA was carried out according to the method described in the work of Borgoyakova et al. [46]. The following reagents were used in ELISA: immunosorbent—recombinant hemagglutinin of influenza virus A/turkey/Stavropol/320-01/2020 (H5N8) [43]; the primary antibodies—sera of immunized animals (dilution from 1:10); the secondary antibodies—goat anti-mouse IgG antibodies labeled with horseradish peroxidase (Sigma-Aldrich, USA); the chromogenic substrate—tetramethylbenzidine (IMTEK, Moscow, Russia); the stop solution—1 N hydrochloric acid solution. Optical density measurements were carried out on a Varioskan LUX device (Thermo Fisher Scientific, USA) at a wavelength of 450 nm. The endpoint titer was determined by the last diluted specimen that gave positive results on the ELISA.

### 2.9. Assessment of T-Cell Immune Response

T-cell immune response was determined using the ELISpot method with the Mouse IFN-gamma ELISpot kit (Mabtech, Stockholm, Sweden). Animal spleens were obtained two weeks after the second immunization. Splenocytes were isolated by sequential homogenization through 70 and 40 μm filters (BD Falcon, New York, NY, USA). After lysis of erythrocytes with lysis buffer (Sigma-Aldrich, USA), splenocytes were washed twice in RPMI medium and placed in 1 mL of RPMI medium with 2 mM L-glutamine and gentamicin (50 μg/mL). Anti-mouse interferon-γ (IFN-γ) antibodies were immobilized on a 96-well plate from the kit. After blocking the plate with RPMI medium with 10% fetal bovine serum, splenocytes were passaged at a rate of 2.5 × 10^5^ cells/well and stimulated with a mixture of virus-specific peptides at a concentration of 20 μg/mL for each peptide (Table 2). The cells were incubated for 20 h at 37 °C in an atmosphere of 5% CO_2_. Then the plates were washed, and biotinylated antibodies against mouse IFN-γ were added and developed using a streptavidin-alkaline phosphatase conjugate and BCIP/NBT substrate. The number of IFN-γ-producing cells was counted using an ELISpot reader (Carl Zeiss, Oberkochen, Germany).

### 2.10. Analysis of Virus-Neutralizing Activity of Immune Sera

Virus neutralization analysis was carried out according to the method described in [43]. At the beginning, the animal sera were treated with Receptor-Destroying Enzyme (RDE) and heat-inactivated prior to the assay. The standardized virus was 200 TCID50/200 µL of viral diluent. Double dilutions of the blood serum were prepared in 200 µL of viral diluent, then 200 µL of the standardized viruses was added into each test tube, and the tubes were incubated for 1 h at 37 °C, 5% CO_2_. Non-immune mouse serum was used as a control. After that, 200 µL suspensions were carried over to the wells of a 96-well plate with MDCK-SIAT1 cell culture; the plates were incubated for 48 h at 37 °C, 5% CO_2_. After incubation, cells were stained with crystal violet solution and analyzed using an Agilent BioTek Cytation 5 multi-mode cell visualization reader (Thermo Fisher Scientific). The serum titer in the virus neutralization test is equal to the serum dilution at which 50% of living cells remain. No more than 5% of living cells remained in the control.

### 2.11. Protectivity Analysis

All studies to assess the protective properties of the vaccine were carried out in compliance with the requirements of SanPiN 3.3686-21 “Sanitary and epidemiological requirements for the prevention of infectious diseases” [47].

A total of 14 days after the second immunization, mice were infected intranasally with 20 MLD50 strain of influenza virus A/Astrakhan/3212/2020 (H5N8). Manipulations to infect mice were carried out using anesthesia with a mixture of tiletamine and zolazepam, and xylazine hydrochloride. Animals were observed after infection daily for 14 days after infection, monitoring clinical signs as indicators of the disease—dishevelment, hypothermia, exhaustion, neurological damage, and death. In cases where mice developed severe conditions incompatible with life, for example, anorexia (loss of >20% of initial body weight), lethargy, the animal was euthanized by cervical dislocation. All other animals were humanely killed by cervical dislocation at the end of the experiment.

### 2.12. Statistics

Statistical data processing was performed using GraphPad Prism 9.0 software (GraphPad Software, Inc., San Diego, CA, USA). Quantitative data are provided as median with range and analyzed using nonparametric tests. The Mann–Whitney test was used to compare two independent groups. Between-group differences were assessed using nonparametric one-way Kruskal–Wallis analysis of variance adjusted for multiple comparisons and Dunn’s statistical hypothesis test. Survival function modeling was performed using the Kaplan–Meier multiplier estimator, and comparison of survival with the control group was performed using the Mantel–Cox test.

## 3. Results

### 3.1. Design and Production of pVAX-H5 Genetic Construct

The gene encoding the HA of the avian influenza A virus (H5N8), which circulated in Stavropol, Russia in 2020 (A/turkey/Stavropol/320-01/2020) among poultry, was selected for the development of a DNA vaccine. The trimerizing domain of bacteriophage T4 fibritin ensuring the formation of hemagglutinin homotrimers was added to the C-terminus of the amino acid sequence of HA [48,49]. In addition, amino acid substitutions were made to the native HA sequence. We relied on previously published work [40] to select the amino acid mutations. To stabilize the HA trimer in the prefusion conformation, the following amino acid residue substitutions were made in the pH switch region of the HA2 subunit. Histidines H26 and H106 were replaced by W (tryptophan) and R (arginine), respectively. K51 and E103 were replaced by isoleucine. To obtain a secreted soluble product, the transmembrane and cytoplasmic domains were removed, and the signal peptide ensuring secretion of the target protein was left unchanged. As a result, the gene sequence encoding the modified hemagglutinin of the influenza A virus (H5N8), named HA/H5, was designed (Figure 1a). In order to assess the ability of the designed amino acid sequence of HA to form trimers, computer modeling of its spatial structure was carried out. According to modeling data, the designed HA sequence is capable of forming specific monomers and trimeric complexes. The model of the designed protein is shown in Figure 1b.

The designed gene was obtained by chemical synthesis and cloned in the pVAX1 plasmid vector. The resulting DNA construct was named pVAX-H5. After that, the DNA vaccine pVAX-H5 was produced and purified, as described in the Materials and Methods section. Using spectrophotometric analysis, it was demonstrated that the plasmid does not contain RNA and genomic DNA impurities, the ratio at wavelengths of 260 and 280 nm was 1.8, and at wavelengths 260 and 230 nm, it was 2.3. Using the LAL test, it was shown that the endotoxin content was no more than 25 EU/dose. The authenticity of the plasmid construct was confirmed by restriction analysis and sequencing.

### 3.2. Analysis of the Modified Hemagglutinin Gene Expression in Eukaryotic Cells Transfected with DNA Vaccine pVAX-H5

The expression of the target gene encoding stabilized hemagglutinin was analyzed by RT-PCR two days after transfection of HEK293 cells with the pVAX-H5 plasmid. To prove that transcription occurs from the target gene, total RNA was isolated from the cells and RT-PCR was performed with specific primers. The results provided in the electropherogram (Figure 2a) have shown that the size of the amplified fragment was 1600 bp, which corresponded to the size of the designed hemagglutinin gene. The results of RT-PCR have shown that specific mRNA was synthesized in HEK293 cells transfected with the constructed plasmid. To analyze the target protein production, Western blot analysis was performed using ferret serum immunized with influenza A (H5N8) virus. The presence of the target protein product was shown both in the cell lysate and in the culture medium (Figure 2b). At the same time, a product was found to be more mobile in the cell lysate than in the culture medium, which may indicate that the protein in the cell was still at the stage of post-translational modifications.

### 3.3. Analysis of the Ability of an Experimental DNA Vaccine to Stimulate the Formation of Specific Humoral and T-Cell Responses

To analyze the immunogenicity of the pVAX-H5 DNA vaccine, BALB/c mice were immunized using jet injection (Figure 3a). The animals were divided into three groups: the first group was injected with pVAX-H5; the second group was the control group, which was injected with the original pVAX1 vector; Group 3 consisted of intact animals.

After double immunization, the animal sera were analyzed using ELISA. Titers of specific antibodies were detected only in the group of animals immunized with pVAX-H5; the average titer was 1 × 10^5^ (Figure 3b).

Sera from immunized mice were also tested for virus neutralization in MDCK cell culture. As a result, it was found that the sera of animals immunized with the pVAX-H5 DNA vaccine are capable of neutralizing live influenza virus strain A/turkey/Stavropol/320-01/2020; the average 50% neutralizing titer was 1 × 10^4^ (Figure 3c).

To assess the ability of the vaccine construct to induce specific cellular immunity, spleens were explanted from mice 14 days after the second immunization, homogenized, and examined using the ELISpot method. The response was assessed by the ability of splenocytes to respond by releasing IFN-γ to stimulation with peptides that were a part of hemagglutinin. Using ELISpot, it was shown that only in the group immunized with the DNA vaccine was specific cellular immunity activated, expressed in a significant increase in the number of lymphocytes producing IFN-γ (Figure 3d).

### 3.4. DNA Vaccine Encoding Modified Hemagglutinin Can Protect Mice from a Lethal Influenza H5N8 Virus Challenge

After the immunogenicity of pVAX-H5 was confirmed, the next stage of our study was to test the ability of the experimental DNA vaccine to stimulate the formation of protective immunity. In these experiments, the activity of the original pVAX-H5 solution and the activity of the lyophilized pVAX-H5 preparation were studied (the lyophilization process is described in Section 2). The following groups were formed. The first group was injected with the original pVAX-H5 (pVAX-H5); the second group was injected with lyophilized pVAX-H5 dissolved in pyrogen-free water (pVAX-H5, lyoph, sucrose); the third group was injected with the inactivated virus A/turkey/Stavropol/320-01/2020 (H5N8) intramuscularly (as a positive control, inactivated H5N8); the fourth group was injected with the original pVAX1 vector (as a negative control, pVAX); the fifth group consisted of intact animals. Immunization was carried out using jet injection. On day 35, blood was collected from the retro-orbital sinus of the animals to assess the level of antibodies after immunization using ELISA. The ELISA results showed that both forms of DNA vaccine induced a specific humoral immune response, without significant differences (*p* > 0.9999, Figure 4b). Specific antibodies were also detected in the group of animals that were injected with the inactivated virus.

On the 35th day after the start of immunization, mice were infected intranasally with the influenza virus A/Astrakhan/3212/2020 (H5N8) (at a dose of 20 MLD50) of the same subclade 2.3.4.4b (for hemagglutinin) as the A/turkey/Stavropol/virus 320-01/2020 (H5N8), but isolated from humans [7]. The results of the experiment (Figure 4c) showed that the survival rate in groups of mice immunized with liquid and lyophilized forms of the DNA vaccine and with the inactivated virus was 100%. In the control groups (non-immunized animals and immunized with the original pVAX1 vector), 10% and 20% of mice survived, respectively (Figure 4c); the surviving mice in the control groups showed clear signs of the disease (hypothermia, decreased activity and appetite), which was not observed in the groups of mice that were immunized with the DNA vaccine prototypes and inactivated virus.

## 4. Discussion

HA is a metastable glycoprotein that undergoes conformational changes after entering the cell and being cut by trypsin into two subunits, HA1 and HA2 [50]. To ensure its immunogenicity, the use of a conformationally correct HA trimer with improved expression and stability for immunization is crucial [51,52]. Previously, in their study, Milder et al. [40] presented data on the search for conserved amino acid residues in the region of pH switches for various strains of influenza virus. The authors showed that the stabilization of recombinant HA can be significantly increased by replacing four amino acid residues (H26W, H106R, K51I, E103I) in pH-sensitive switches involved in protein refolding. This recovery and stabilization approach was found to be applicable to influenza A HA groups 1 and 2. In our study, we relied on data from Milder et al. [40] during the design of the HA sequence of the avian influenza A virus (H5N8) (A/turkey/Stavropol/320-01/2020). In the pH switch region of the HA2 subunit, amino acid residues were replaced (H26W, H106R, K51I, E103I), and the fibritin domain of bacteriophage T4 was additionally added to the C-terminus to form HA homotrimers. Also, the transmembrane and cytoplasmic domains were removed from the sequence of the original hemagglutinin, and the signal peptide was left unchanged to obtain a secreted soluble product. Modeling of the spatial structure of the designed HA showed the possibility of forming specific monomers and trimers (Figure 1a). The AlphaFold bioinformatics tool allowed us to predict the structure of a protein molecule [53]. However, this is a predictive tool [54] and it is necessary to obtain this recombinant protein and study its physicochemical and structural features for a more accurate understanding of its characteristics and stability; we plan to carry out this research in further studies.

The gene based on the designed amino acid sequence of HA was synthesized and cloned in pVAX1 plasmid vector to obtain the DNA vaccine pVAX-H5. Target gene expression in eukaryotic cells transfected with pVAX-H5 was studied using two methods; transcription was confirmed by detection of specific mRNAs using RT-PCR (Figure 2a), and target protein synthesis was confirmed using Western blot analysis (Figure 2b). The identification of specific mRNA suggests that pVAX-H5 is efficiently transcribed in mammalian cells. Identification of the target protein product in the culture medium in Western blot analysis indicates that the product is secreted from the cell; the protein secretion was achieved by designing a targeted immunogen, namely by retaining the natural signal peptide and removing the transmembrane and cytoplasmic domains [43,55].

One of the disadvantages of DNA vaccines is their low immunogenicity when administered as a naked plasmid DNA. This problem may be solved by the use of modern NFIS. An increase in the immunogenicity of a DNA vaccine administered using NFIS may be associated with a wider distribution of the injected drug within tissues and more efficient delivery into cells [11,14,31]. Moreover, the injection process can act as a physical adjuvant during vaccination [56]. An assessment of the immunogenicity of the resulting DNA vaccine showed that immunization of BALB/c mice using NFIS induced high titer of specific antibodies with virus-neutralizing activity (Figure 3). The presence of specific antibodies with virus-neutralizing activity may indirectly indicate that the HA protein, the synthesis of which is ensured by pVAX-H5, is presented to the immune system in the correct conformation [51,52].

The undeniable advantage of DNA vaccines is the ability to induce a balanced B- and T-cell immune response [19,20]. It has been shown that immunization with pVAX-H5 elicits a specific T-cell response. Although T cells cannot provide sterilizing immunity against influenza, they can often provide broader protection against different influenza strains or subtypes [32].

In order to be able to store DNA vaccine preparations for a long time at a temperature above zero and to increase its storage period, a solution of the pVAX-H5 plasmid in sucrose (2 mg/mL) was freeze-dried. After lyophilization, the immunogenicity and protectivity of the original and lyophilized forms of the DNA vaccine were assessed. It has been shown that immunization of BALB/c mice with both liquid and lyophilized forms of pVAX-H5 induced a high level of synthesis of virus-specific antibodies, and provided 100% protection from lethal infection with influenza A virus (H5N8) (Figure 4). In this study, the first analysis of the immunogenicity of the resulting experimental DNA vaccine was carried out. In further work, it is necessary to investigate the ability of the vaccine to provide protection against heterologous strains and the dose-dependent effect. It will also be important to confirm the protective properties of the DNA vaccine pVAX-H5 in other animal models, such as ferrets.

## 5. Conclusions

Thus, in this study, an experimental DNA vaccine pVAX-H5 encoding a modified hemagglutinin of the influenza A/Astrakhan/3212/2020 (H5N8) virus was obtained. It has been shown that immunization of BALB/c mice using jet injection with the pVAX-H5 plasmid induced the synthesis of specific antibodies with virus-neutralizing activity and elicited a cellular immune response. It has been shown that after lyophilization of pVAX-H5 in sucrose solution, the plasmid retained its biological activity. Both liquid and lyophilized forms of pVAX-H5 DNA vaccine provided 100% protection of immunized mice against lethal challenge with influenza A virus A/turkey/Stavropol/320-01/2020 (H5N8). The results of the study indicate that pVAX-H5 has good opportunities as a vaccine candidate against the influenza A virus (H5N8).

## Figures and Tables

**Figure 1 vaccines-12-00538-f001:**
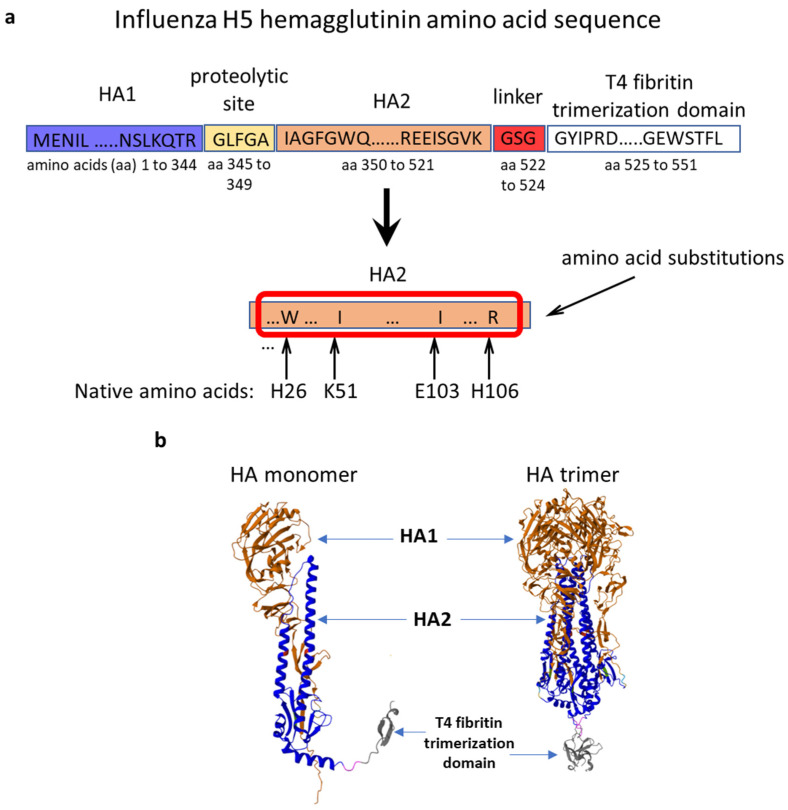
Design of the modified HA amino acid sequence. (**a**) Scheme for designing the amino acid sequence of influenza A virus (H5N8) hemagglutinin. (**b**) Modeling the structure of influenza A virus (H5N8) hemagglutinin. The different structural elements are shown in different colors: HA1 subunit (light brown), HA2 subunit (blue), linker (rose) and T4 fibritin trimerization domain (grey).

**Figure 2 vaccines-12-00538-f002:**
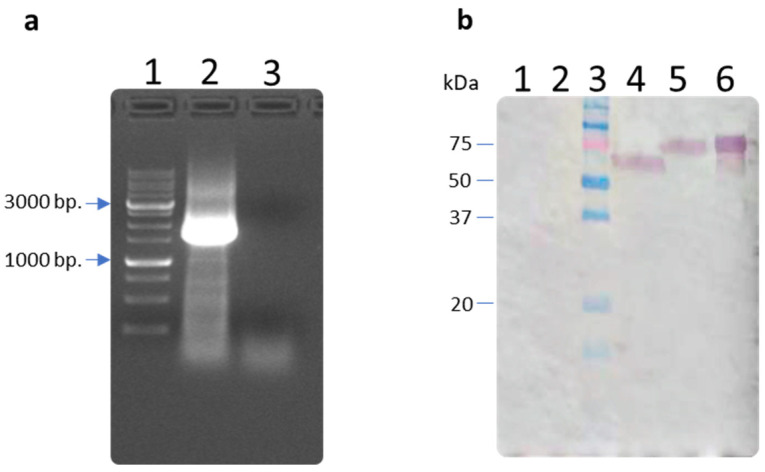
Expression of the hemagglutinin gene. (**a**) Electropherogram of RT-PCR products in 1% agarose gel: lane 1—1 kb DNA ladder (SibEnzyme); lane 2—use of total RNA pVAX-H5 as a template; lane 3—use of total RNA from pure cells as a template; (**b**) analysis of HA/H5 protein production in HEK293 cells by Western blotting using serum of a ferret immunized with influenza A (H5N8) virus: Lane 1—lysate of untransfected HEK293 cells; Lane 2—culture medium collected from untransfected HEK293 cells; Lane 3—molecular weight marker Precision Plus Protein Dual Color Standards (Bio-rad, Hercules, CA, USA); Lane 4—lysate of HEK293 cells transfected with plasmid pVAX-H5; Lane 5—culture medium collected from HEK293 cells transfected with the pVAX-H5 plasmid; Lane 6—recombinant hemagglutinin HA/H5 of influenza A virus (H5N8) (K+) [43].

**Figure 3 vaccines-12-00538-f003:**
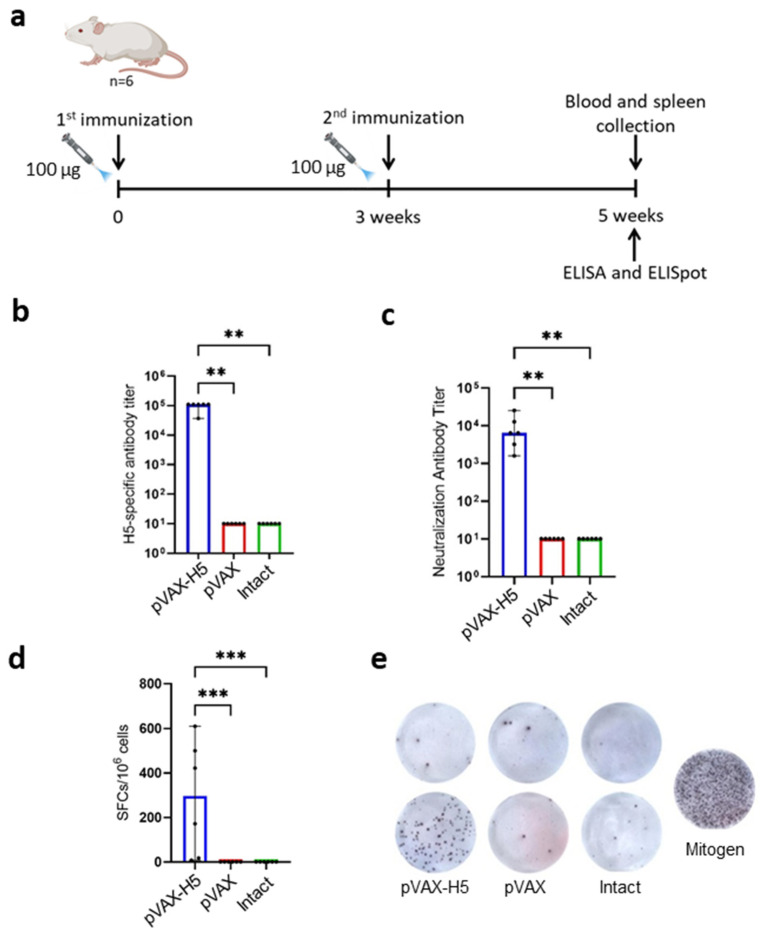
Analysis of humoral and cellular response to DNA vaccine pVAX-H5. (**a**) Scheme of mouse immunization. (**b**) Reciprocal titer (dilution) of hemagglutinin-specific antibodies detected in immune sera using ELISA. (**c**) The neutralization activity of immune sera against influenza virus A/turkey/Stavropol/320-01/2020 (H5N8). Reciprocal titer values are provided in the plot. (**d**) ELISpot results. Number of splenocytes expressing IFN-γ in response to stimulation with a pool of specific peptides, determined by ELISpot. Each bar represents the average number of IFN-γ spot-forming cells (SFCs) per million splenocytes stimulated. (**e**) The representative images of ELISpot wells (top row: splenocytes not stimulated with peptides; bottom row: splenocytes stimulated with peptide pool or a mitogen). pVAX-H5—a group of animals immunized with the liquid form of the pVAX-H5 DNA vaccine; pVAX—group of animals immunized with the pVAX1 vector; intact—a group of animals that have not been subjected to any manipulation. In panels (**b**–**d**), data are provided as median with range. Significance was calculated using non-parametric one-way Kruskal–Wallis analysis of variance with correction for multiple comparisons and Dunn’s statistical hypothesis test (** *p* < 0.01, *** *p* < 0.001).

**Figure 4 vaccines-12-00538-f004:**
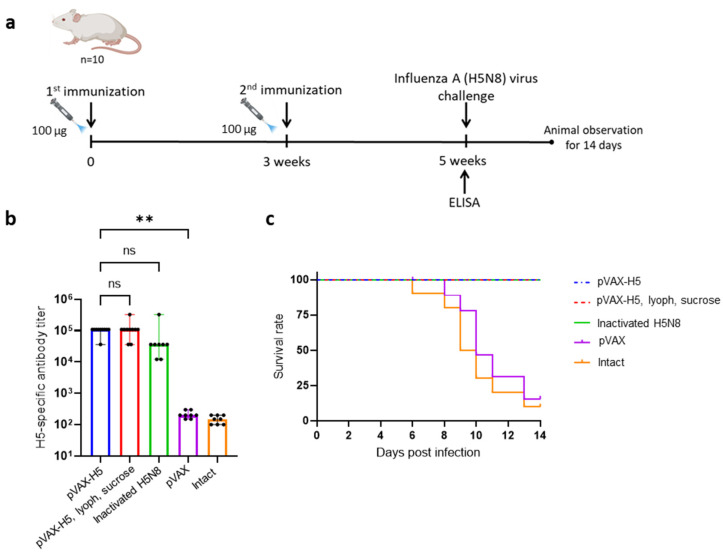
Immunogenic and protective properties of DNA vaccine pVAX-H5. (**a**) Scheme of mice immunization and virus challenge. (**b**) ELISA results. Reciprocal titer values are provided in the plot. Data are provided as median with range. Significance was calculated using non-parametric one-way Kruskal–Wallis analysis of variance with correction for multiple comparisons and Dunn’s statistical hypothesis test (** *p* < 0.01, ns: no significance). (**c**) Survival curves of immunized animals after infection with the influenza virus strain A/Astrakhan/3212/2020 (H5N8). Survival function modeling was performed using the Kaplan–Meier multiplier estimator, and comparison of survival with the control group was performed using the Mantel–Cox test. pVAX-H5—a group of animals immunized with the liquid form of the pVAX-H5 DNA vaccine; pVAX-H5, lyoph., sucrose—a group of animals immunized with a lyophilized form of DNA vaccine pVAX-H5; inactivated H5N8—a group of animals immunized with the inactivated H5N8 virus; pVAX—group of animals immunized with the pVAX1 vector; intact—a group of animals that have not been subjected to any manipulation.

**Table 1 vaccines-12-00538-t001:** Primers for RT-PCR.

Primer	Nucleotide Sequence (5′-3′)
forward	TTTCTGGCTAGCGCCGCCACCATGGAGAACA
reverse	AAAAAAAGCGGCCGCTCATTACAGGAAGGT

**Table 2 vaccines-12-00538-t002:** Peptides from the hemagglutinin of the influenza A/H5N8 virus used to stimulate splenocytes *.

№	Peptides from the H5 Protein
1	MPFHNIHPL
2	AGWLLGNPM
3	CYPGSLNDY
4	RVPEWSYIV
5	LRNSPLREKRRKRGL
6	YVKSNKLVL

* The peptides listed in the table have a high probability of binding affinity to MHC H-2D mice BALB/c.

## Data Availability

The data can be shared up on request.

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
