# Peer review of "DNA Vaccine Encoding a Modified Hemagglutinin Trimer of Avian Influenza A Virus H5N8 Protects Mice from Viral Challenge"

_vaccines, 2024, doi:10.3390/vaccines12050538_

Round 1
Reviewer 1 Report
Comments and Suggestions for Authors
In this manuscript, Litvinova et al. formulated a DNA vaccine encoding hemagglutinin trimer of H5N8 avian influenza virus and tested its immunogenicity and protection in a mouse model. There are several areas to improve in this manuscript:
Major comments:
1. The animal experiment needs clarity. Did the authors monitor body weight change or temperature change? Clinical signs are a very subjective measure and not reliable. Was ‘found dead’ used as the endpoint for the experiment? What was the humane endpoint otherwise? In typical influenza experiments in a mouse model, a loss in body mass between 20-30% is considered as humane endpoint from animal welfare perspective. It is not clear what the criteria were, as approved by their animal care and use committee. Protection during virus challenge is also supported by measurement of lung virus titers at different days post infection (challenge). Did the authors look for virus titers?
2. The discussion section does not contain any discussion at all. It needs to be completely rewritten. Lines 392 to 412 are already repeated from the introduction section. Lines 427 to 442 are repetition of results. There is nothing discussed about their result and how compares with other findings, its relevance, limitations, future perspectives etc.
Other comments:
1. In introduction, in lines 34-36, sentence is not clear.
2. Define cells and abbreviations in their first use.
3. The purpose of modeling was not clear.
4. Provide more details about animal inhalation anesthesia used during vaccination. Is it isoflurane?
5. In first experiment, was the vaccine used a solution or lyophilized form?
6. For virus neutralization assay, was 48hr enough to get cell sloughing?
7. Write the active ingredients of the anesthesia used during virus infection.
8. Lines 366-367 how was the ‘physiological reactions’ measured on mice after infection.
9. In figure 4a, add 2 weeks follow up period as well.
10. Keep consistency in use of same denotation for the vaccine group. There is discrepancy in text, figures, and figure legends.
11. In figure legends, one set of description for vaccine groups is enough for one figure. No need to repeat the same description but keep consistency in denotation between figures and legends.
12. Line 381, mean+/-a.d. What is a.d.?
Author Response
We would like to thank the reviewer for careful and thorough reading of our manuscript and for the thoughtful comments and constructive suggestions, which help to improve its quality. Our response follows.
Corrections to the text are highlighted in red.
In this manuscript, Litvinova et al. formulated a DNA vaccine encoding hemagglutinin trimer of H5N8 avian influenza virus and tested its immunogenicity and protection in a mouse model. There are several areas to improve in this manuscript:
Major comments:
- The animal experiment needs clarity. Did the authors monitor body weight change or temperature change? Clinical signs are a very subjective measure and not reliable. Was ‘found dead’ used as the endpoint for the experiment? What was the humane endpoint otherwise? In typical influenza experiments in a mouse model, a loss in body mass between 20-30% is considered as humane endpoint from animal welfare perspective. It is not clear what the criteria were, as approved by their animal care and use committee. Protection during virus challenge is also supported by measurement of lung virus titers at different days post infection (challenge). Did the authors look for virus titers?
Dear reviewer, in our study, all experiments with animals were carried out in accordance with the principles of humanity and were regulated by international and state rules. If, as a result of the disease, the animal developed a serious condition incompatible with life: anorexia (loss of >20% of the initial body weight), the animal was euthanized by cervical dislocation. Thus, we indicated anorexia (loss of >20% of initial body weight) and subsequent euthanasia as death.
We have made the relevant clarifications and corrections.
We agree that isolation of the virus from the immunized animals and analysis of the transmissibility of the virus is important for assessing the effectiveness of the vaccine. However, virus titers after infection were not determined, since we gave a lethal dose of the virus, so the absence of death was an indicator of protection. But we will try to carry out this analysis at the next stages of the study of our vaccine. At the first stage of the study, it was important for us to evaluate the immunogenic and protective properties of the experimental DNA vaccine.
- The discussion section does not contain any discussion at all. It needs to be completely rewritten. Lines 392 to 412 are already repeated from the introduction section. Lines 427 to 442 are repetition of results. There is nothing discussed about their result and how compares with other findings, its relevance, limitations, future perspectives etc.
We have revised the text to address the reviewer’s concern and we hope that it is clearer now.
Other comments:
- In introduction, in lines 34-36, sentence is not clear.
The correction has been made. Please see page 1 of the revised manuscript, lines 34-36.
In 2020, cases of human infection with the influenza A virus (H5N8) were identified; later this strain was recommended by WHO as a vaccine strain [6, 7].
- Define cells and abbreviations in their first use.
The correction has been made. Please see page 2 of the revised manuscript, lines 93-95.
Human Embryonic Kidney 293 (HEK293) and Madin-Darby canine kidney (MDCK) cell cultures (Cell Culture Collection of FBRI SRC VB «Vector», Rospotrebnadzor) were used in the study.
- The purpose of modeling was not clear.
The correction has been made. Please see page 6 of the revised manuscript, lines 277-280.
In order to assess the ability of the designed amino acid sequence of HA to form trimers, computer modeling of its spatial structure was carried out. According to modeling data, the designed HA sequence is capable of forming specific monomers and trimeric complexes
- Provide more details about animal inhalation anesthesia used during vaccination. Is it isoflurane?
The correction has been made. Please see page 4 of the revised manuscript, lines 162-173.
For animal immunization we used a gas anesthesia unit (RWD Life Science, China) using 2.5% isoflurane. We are familiar with the likely detrimental effects of this gas on the structure of the lungs and therefore the exposure time was minimal and was about 4-5 minutes.
In the section “Immunization of BALB/c mice with DNA vaccine pVAX-H5” the following sentence was added - To immobilize the animals, inhalation anesthesia (RWD Life Science, China) with a 2.5% isoflurane solution was used for 4-5 minutes.
- In first experiment, was the vaccine used a solution or lyophilized form?
In the first experiment, the vaccine solution in saline was used.
The correction has been made. Please see page 4 of the revised manuscript, line 177.
- For virus neutralization assay, was 48hr enough to get cell sloughing?
This time is sufficient for virus neutralization analysis. A clarification in the Materials and Methods section has been made.
- Write the active ingredients of the anesthesia used during virus infection.
Manipulations to infect mice were carried out using anesthesia with a mixture of Zoletil 100 and Xila.
The correction has been made. Please see page 6 of the revised manuscript, lines 246-247.
- Lines 366-367 how was the ‘physiological reactions’ measured on mice after infection.
We apologize for our error.
The correction has been made. Please see page 6 of the revised manuscript, lines 247-250.
Animals were observed after infection daily for 14 days after infection, monitoring clinical signs as indicators of the disease – dishevelment, hypothermia, exhaustion, neurological damage and death.
- In figure 4a, add 2 weeks follow up period as well.
The correction has been made. Please see page 10 of the revised manuscript, line 386.
- Keep consistency in use of same denotation for the vaccine group. There is discrepancy in text, figures, and figure legends.
We apologize for our error.
The correction has been made. Please see page 10 of the revised manuscript, lines 363-370.
- In figure legends, one set of description for vaccine groups is enough for one figure. No need to repeat the same description but keep consistency in denotation between figures and legends.
The correction has been made. Please see page 11 of the revised manuscript, lines 387-389.
- Line 381, mean+/-a.d. What is a.d.?
We apologize for our error. The correction has been made. P Please see page 11 of the revised manuscript, lines 387-389.
Reviewer 2 Report
Comments and Suggestions for Authors
In this study, Litvinova et al. developed an experimental DNA vaccine called pVAX-H5, which encodes a modified hemagglutinin of the influenza A/Astrakhan/3212/2020 (H5N8) virus. The study demonstrated that both liquid and lyophilized forms of the pVAX-H5 DNA vaccine provided 100% protection to immunized mice against a lethal challenge with the influenza A virus. These results suggest that pVAX-H5 has potential as a candidate vaccine against the influenza A virus. However, several major concerns should be addressed and clarified.
Major issues:
1. What antibodies are used in Fig 2B? Is HA-specific antibody used for detection?
2. The number of spots in Figure 3E needs to be statistically analyzed.
3. Please note the dosage of the vaccine for Figure 3A and 4A.
4. Does the immunization with pVAX-H5 have an impact on the body weight of mice?
5. Why was the viral load in various organs not tested after the challenge?
Minor issues:
1. The name of the gene needs to be italicized.
2. The introduction lacks an overview of the research progress on AIV DNA vaccines.
3. The figure lacks a subtitle.
Author Response
We would like to thank the reviewer for careful and thorough reading of our manuscript and for the thoughtful comments and constructive suggestions, which help to improve its quality. Our response follows.
Corrections to the text are highlighted in red.
In this study, Litvinova et al. developed an experimental DNA vaccine called pVAX-H5, which encodes a modified hemagglutinin of the influenza A/Astrakhan/3212/2020 (H5N8) virus. The study demonstrated that both liquid and lyophilized forms of the pVAX-H5 DNA vaccine provided 100% protection to immunized mice against a lethal challenge with the influenza A virus. These results suggest that pVAX-H5 has potential as a candidate vaccine against the influenza A virus. However, several major concerns should be addressed and clarified.
Major issues:
- What antibodies are used in Fig 2B? Is HA-specific antibody used for detection?
The correction has been made. Please see page 8 of the revised manuscript, lines 305-306.
Analysis of HA/H5 protein production in HEK293 cells was carried out by western blotting using serum of a ferret immunized with influenza A (H5N8).
- The number of spots in Figure 3E needs to be statistically analyzed.
Figure 3E shows representative spots, statistical analysis is provided in Figure 3d.
- Please note the dosage of the vaccine for Figure 3A and 4A.
The correction has been made. Please see Figure 3A and 4A of the revised manuscript.
- Does the immunization with pVAX-H5 have an impact on the body weight of mice?
In this work, we did not evaluate the effect of pVAX-H5 on body weight, but outwardly the mice did not differ from the control.
- Why was the viral load in various organs not tested after the challenge?
We agree that isolation of the virus from the immunized animals and analysis of the transmissibility of the virus is important for assessing the effectiveness of the vaccine. However, virus titers after infection were not determined, since we gave a lethal dose of the virus, so the absence of death was an indicator of protection. But we will try to carry out this analysis at the next stages of the study of our vaccine. At the first stage of the study, it was important for us to evaluate the immunogenic and protective properties of the experimental DNA vaccine.
Minor issues:
- The name of the gene needs to be italicized.
The correction has been made. Please see page 6 of the revised manuscript, line 277.
- The introduction lacks an overview of the research progress on AIV DNA vaccines.
The correction has been made. Please see page 2 of the revised manuscript, lines 52-59.
Many candidate preventive DNA vaccines against human viral and bacterial diseases are at the stage of clinical trials [16], as well as other vaccines at various stages of clinical trials, show that DNA vaccines are safe and have an acceptable reactogenicity profile [16, 17, 18]. DNA vaccines against AIV are being developed [19, 20]. A DNA vaccine encoding hemagglutinin of the influenza A virus (H5N1) successfully passed the phase I of clinical trials [21]. It should also be noted that the DNA vaccine against highly pathogenic influenza A (H5N1) for chicken immunization, recently licensed by the US Department of Agriculture, was developed by Agrilabs [22].
- The figure lacks a subtitle.
The correction has been made. Please see Figure 1, 2, 3 and 4 of the revised manuscript.
Round 2
Reviewer 1 Report
Comments and Suggestions for Authors
Most of the comments are addressed. The details about animal monitoring and loss of >20% body weight as the measure of humane endpoint is mentioned on the response to reviewer but not in the manuscript. Likewise, for the anesthesia is it Ketamine and Xylazine? Rather than the commercial name on bottles, use the active ingredient names like ketamine and xylazine.
Comments on the Quality of English LanguageMinor English edit is recommended.
Author Response
We would like to thank the reviewer for careful and thorough reading of our manuscript and for the thoughtful comments and constructive suggestions, which help to improve its quality.
The correction has been made. Please see page 6 of the revised manuscript, subparagraph «Protectivity analysis».